# Synergistic Photothermal Therapy and Chemotherapy Enabled by Tumor Microenvironment-Responsive Targeted SWCNT Delivery

**DOI:** 10.3390/ijms25179177

**Published:** 2024-08-23

**Authors:** Shuoye Yang, Jiaxin Liu, Huajian Yuan, Qianqian Cheng, Weiwei Shen, Yanteng Lv, Yongmei Xiao, Lu Zhang, Peng Li

**Affiliations:** 1School of Biological Engineering, Henan University of Technology, Zhengzhou 450001, China231060400119@stu.haut.edu.cn (Q.C.); 18338165651@163.com (W.S.); 18625642082@163.com (Y.L.); zhanglu@haut.edu.cn (L.Z.); 2Institute for Complexity Science, Henan University of Technology, Zhengzhou 450001, China; 3School of Chemistry and Chemical Engineering, Henan University of Technology, Zhengzhou 450001, China; xym9510@126.com

**Keywords:** photothermal therapy, SWCNTs, synergistic therapy, targeted delivery, tumor microenvironment response

## Abstract

As a novel therapeutic approach, photothermal therapy (PTT) combined with chemotherapy can synergistically produce antitumor effects. Herein, dithiodipropionic acid (DTDP) was used as a donor of disulfide bonds sensitive to the tumor microenvironment for establishing chemical bonding between the photosensitizer indocyanine green amino (ICG-NH_2_) and acidified single-walled carbon nanotubes (CNTs). The CNT surface was then coated with conjugates (HD) formed by the targeted modifier hyaluronic acid (HA) and 1,2-tetragacylphosphatidyl ethanolamine (DMPE). After doxorubicin hydrochloride (DOX), used as the model drug, was loaded by CNT carriers, functional nano-delivery systems (HD/CNTs-SS-ICG@DOX) were developed. Nanosystems can effectively induce tumor cell (MCF-7) death in vitro by accelerating cell apoptosis, affecting cell cycle distribution and reactive oxygen species (ROS) production. The in vivo antitumor activity results in tumor-bearing model mice, further verifying that HD/CNTs-SS-ICG@DOX inhibited tumor growth most significantly by mediating a synergistic effect between chemotherapy and PTT, while various functional nanosystems have shown good biological tissue safety. In conclusion, the composite CNT delivery systems developed in this study possess the features of high biocompatibility, targeted delivery, and responsive drug release, and can achieve the efficient coordination of chemotherapy and PTT, with broad application prospects in cancer treatment.

## 1. Introduction

Photothermal therapy (PTT) uses photothermal conversion agents (PTAs) to obtain energy from near-infrared (NIR) light [1], which is converted into heat to increase the temperature of the surrounding environment to approximately 46 °C, resulting in tumor cell death and tumor necrosis [2]. Most chromophores inherent in the body, such as oxygenated hemoglobin, deoxygenated hemoglobin, and melanin, are relatively weakly absorbed in the range of NIR light, allowing it to penetrate deep tissue and thus be highly efficient for PTT [3,4]. However, PTT always leads to poor efficacy when used alone due to damage to the normal tissue around the tumor, the limited depth of light penetration leading to the incomplete ablation of tumors outside the irradiation range, and difficulties in the delivery of PTA. Therefore, PTT must be combined with other treatments to compensate for its inherent shortcomings.

Despite killing tumor cells and inhibiting their growth and reproduction, chemotherapy inevitably produces a certain degree of toxicity to normal human cells [5], especially those with rapid division and proliferation. Heat shock proteins (HSPs) are overexpressed in tumor tissues, which participate in protein folding and enhance protein stability, promote the proliferation and metastasis of tumor cells, maintain cell survival through various ways, and inhibit cell apoptosis. Synergistic therapy (NIR irradiation) can inhibit the expression of HSP, thereby improving the efficacy. Combining PTT and chemotherapy can reduce the required dose of chemotherapeutic drugs and decrease adverse reactions [6]. When NIR radiation is applied to the tumor site, the photothermal effect will generate heat energy, and the increase in temperature can promote the release of chemotherapy drugs from carriers to increase their accumulation in the tumor site, thus realizing the effective combination of PTT and chemotherapy [7]. Therefore, the combination of PTT and chemotherapy could be a promising cancer treatment strategy. Previously, graphene oxide, indocyanine green (ICG), and doxorubicin hydrochloride (DOX) nanoparticle cores were coated with erythrocyte membrane and folic acid [8]. The system combined PTT with chemotherapy to effectively reduce tumor size. Thus, PTT combined with chemotherapy was shown to be more effective than either PTT or chemotherapy alone. However, because of their strong phototoxicity, low dark toxicity (low cytotoxicity under dark conditions) [9], and rapid clearance by the liver and kidney, it is still challenging to load PTAs, especially ICG, onto delivery platforms such as single-walled carbon nanotubes (SWCNTs) to form a vector with controlled release.

Nanomaterials commonly used as delivery carriers of chemotherapeutic drugs and PTAs include carbon nanotubes (CNTs), quantum dots, gold nanoparticles, and gold nanorods, among others. CNTs have the advantages of high chemical stability, large specific surface area, good drug adsorption, and ease of functionalization [10]. Among them, SWCNTs have garnered extensive application in the development of versatile nanoplatforms because of their remarkable photothermal conversion ability and sizable drug loading capacity. Furthermore, the strong NIR absorption of SWCNTs makes it an effective absorbent for the photothermal ablation of cancer cells [11,12]. SWCNTs have an adjustable hollow chemical structure, and the drug can be loaded onto their surface through covalent bonds or π–π interaction and the adsorption complexation of charged surfactants [13]. The surface functional modification of SWCNTs improves their dispersion, cell uptake, and intratumoral aggregation while reducing their cytotoxicity [14]. Although SWCNTs have already been applied in some studies for synergistic tumor treatment through PTT coupled with chemotherapy, the poor targeting delivery of SWCNT carriers needs to be further addressed [15].

Disulfide bonds exist naturally in the human body and play an important role in stabilizing protein structures [16]. Glutathione (GSH) has the ability to convert disulfide bonds into sulfhydryl groups. When ICG is conjugated onto the surface of a nanocarrier by a disulfide bond, the nanocarrier can rapidly release ICG under a tumor microenvironment (TME). This modification can enhance the hydrophilicity and stability of ICG; meanwhile, the photothermal conversion performance of the nanocarrier is also improved. GSH concentrations in tumor cells are high; therefore, SWCNTs and ICG can be grafted by disulfide bonds to form a drug delivery carrier that is sensitive to TME. Responsive and “intelligent” drug delivery systems (DDSs) provide a new mechanism to control cargo (such as ICG) release through internal or external stimuli [17]. Adaptation to internal oxidative stress in tumor cells usually leads to an increase in the antioxidant capacity of the surrounding tumor tissue, causing selective internal stimulation to trigger the responsive release of ICG, thus avoiding their photolysis and premature release and maximizing the photothermal effect at the tumor site [18]. Redox-sensitive delivery carriers that contain a disulfide bond are widely used in intracellular drug delivery towards tumors due to degradation from the sulfhydryl exchange reaction in the reducing environment of tumors [19]. Some studies have reported the use of disulfide bonds to control drug release. For example, camptothecin (CPT) was linked to polyethylene glycol (PEG) to design a CPT-SS-PEG-SS-CPT delivery carrier, which releases the drug into TME, improving the antitumor effect and reducing the side effects of drugs.

In our previous study, we showed that a conjugate (HD) of hyaluronic acid (HA) and a 1,2-tetragacylphosphatidyl ethanolamine coating can increase nanosystems’ targeting affinity to tumors. Herein, Dithiodipropionic anhydride (DTDPA) was synthesized by the dehydration of dithiodipropionic acid (DTDP), which provides a disulfide bond connecting CNTs and ICG. A HD-coated SWCNT delivery system was further developed to load the chemotherapeutic drug, DOX. The DOX and ICG co-loaded system (HD/CNTs-SS-ICG@DOX) can responsively release ICG under TME and achieve synergistic antitumor effects by mediating chemotherapy and PTT [20]. After HD/CNTs-SS-ICG@DOX reach the tumor site by the targeted action of HA, they first penetrate the cell membrane and reach the lysosome through endocytosis. At this stage, high concentrations of GSH in TME will break disulfide bonds, thus rapidly releasing ICG. At the same time, external NIR light will be used to convert light energy into high-temperature thermal energy to maximize the photothermal effect of ICG. The acidic microenvironment and the increase in temperature promote the release of DOX from CNTs, and then lysosomal escape occurs, which makes small molecules enter the cytoplasm and nucleus, there realizing an effective coordination between PTT and chemotherapy. This tumor-targeting and microenvironment-responsive nanosystem has broad application potential for PTT synergistic chemotherapy.

## 2. Results and Discussion

### 2.1. Characterization of Various SWCNT Nanocarriers

The preparation procedure and antitumor mechanism of HD/CNTs-SS-ICG@DOX are shown in Figure 1 and Appendix A. To improve the biocompatibility, pristine SWCNTs were treated with nitric acid to obtain CNTs with carboxyl groups on their surface [21]. After acidification, the particle size of CNTs decreased from 164.2 nm to approximately 122.4 nm (Figure 1C and Table 1). The results listed in Table 2 show that the melting point of DTDP was in the range of 152.5–155.8 °C, and that of DTDPA was in the range of 67.5–73.2 °C. According to reports in the literature [5], the melting point of DTDP is 153–155 °C, and the change in melting point confirmed the successful preparation of DTDPA. TEM images showed that the acidified CNTs had a larger diameter, stretched length, and improved dispersity, with the removal of most impurities (Figure 1A). Despite the various treatments, SWCNT samples maintained a hollow tubular structure, indicating that the addition of each modified component did not destroy its basic structure. Moreover, numerous nodules appearing on the surface of HD/CNTs-SS indicated that HD was conjugated onto CNT samples. SEM images of different SWCNTs are shown in Appendix A.

The characteristic peaks of DTDP, DTDPA, NH_2_-PEG-OH, and NH_2_-PEG-SS-COOH were assessed using a nuclear magnetic resonance hydrogen spectrometer (Figure 1B). DTDP and DTDPA had the same two peaks at δ = 2.88 ppm (-CH_2_-SS-CH_2_-) and δ = 2.60 ppm (-CH_2_-CH_2_-SS-CH_2_-CH_2_-), respectively, and the carboxyl hydrogen in DTDPA disappeared after the dehydration of DTDP to DTDPA. The carboxyl hydrogen at δ = 12.36 ppm (-COOH) disappeared in DTDPA. The peaks at δ = 2.94 ppm (-CH_2_-SS-CH_2_-) and δ = 2.83 ppm (-OOC-CH_2_-CH_2_-SS-CH_2_-CH_2_-COOH) showed the successful synthesis of NH_2_-PEG-SS-COOH [5]. The FT-IR spectra in Figure 1E show that DTDP had the characteristic peaks of C=O and -OH at 3400 cm^−1^ and 1600–1700 cm^−1^, whereas DTDPA only had a vibration peak at 1689 cm^−1^, indicating that DTDP had been completely dehydrated to DTDPA. NH_2_-PEG-SS-COOH had characteristic peaks of C=O and -OH, respectively. Compared with CNT, after conjugation with NH_2_-PEG-SS-COOH, the C=O peak of the CNTs-SS became weak and the absorption peaks from the carboxyl group and the disulfide bond were notably enhanced. These results verified the successful synthesis of NH_2_-PEG-SS-COOH and surface modification for CNT samples.

The efficient loading of ICG and DOX by CNT carriers was demonstrated by ultraviolet-visible spectrophotometry (UV-Vis) (Figure 1F). HD/CNTs-SS-ICG@DOX showed intense UV absorption peaks from DOX and ICG at 480 nm and 780 nm, respectively, indicating the presence of DOX and ICG on CNT carriers. The zeta potential analysis results indicated that, compared with the pristine SWCNTs, the surface charge of CNTs decreased from −21.8 mV to −41.7 mV due to the large number of carboxyl groups on their surface (Figure 1D and Table 1). After modification with NH_2_-PEG-SS-COOH, an increase in the electric potential to −18.5 mV for CNTs-SS was observed, attributing the neutralization of the carboxyl group during the amide reaction and its strong adsorption capacity. Finally, HD/CNTs-SS-ICG@DOX had the potential of −18.5 mV. In Raman spectra, the D peak at 1340 cm^−1^ and the G peak at 1575 cm^−1^ represented the disordered vibration and stretching vibration of the C-C single bond, respectively, indicating that the modified CNTs retained their basic structure (Figure 1G). After modification with NH_2_-PEG-SS-COOH, the I_G_/I_D_ value increased, indicating an increase in the degree of graphitization. The enhancement of the G peak was mainly caused by the enlarged diameter of CNTs and the enhanced stretching vibration of the C-C single bond. For HD/CNTs-SS, their I_G_/I_D_ value further increased from 3.2392 to 3.8246, which was likely attributed to the strong interaction between the phosphatidyl chains in HD and SWCNTs, resulting in a change in the force constants of C-C single bonds in the wall structure. XRD patterns showed that the most typical diffraction peaks at 26° and 44° indicated that the modified CNTs still retained their basic structure (Figure 1H).

Additionally, the dispersion stability of SWCNTs@DOX and HD/CNTs-SS@DOX are shown in Appendix A. Regardless of which dispersion medium was used, SWCNTs@DOX always tended to be rapidly clustered. In contrast, HD/CNTs-SS@DOX exhibited better dispersion stability in various mediums. In X-ray photoelectron spectroscopy (XPS) spectra, CNTs-SS and HD/CNTs-SS showed the N1s peaks at 400 eV, and only HD/CNTs-SS had the characteristic P peaks at 132 eV and 188 eV, indicating that NH_2_-PEG-SS-COOH and HD were conjugated onto the surface of CNTs (Appendix A). The thermogravimetric analysis (DSC) results showed that SWCNTs transformed from an amorphous to crystalline state at 252.56–264.09 °C, while the temperature range for acidified CNTs to transform was 107.87–161.91 °C. The significant decrease in phase transition temperature was caused by the thermal decomposition of the hydroxyl and carboxyl groups on the surface of CNTs into CO_2_ and H_2_O (Appendix A). The DSC curves of CNTs-SS and HD/CNTs-SS were similar, and the decomposition temperature was higher than that of CNTs, indicating that modification with NH_2_-PEG-SS-COOH and HD could increase the thermal stability of CNTs.

### 2.2. Loading Efficiency and In Vitro Release of ICG and DOX

The ICG loading efficiencies of CNTs-SS-ICG and HD/CNTs-SS-ICG were 90.64% and 88.43%, respectively. The DOX encapsulation efficiency and loading efficiency of various delivery systems are listed in Table 3. For example, the encapsulation efficiency and loading efficiency of HD/CNTs-SS-ICG@DOX were 88.40% and 55.82 μg/mg, respectively.

To assess the redox responsiveness of the nanosystem, the release of ICG from HD/CNTs-SS-ICG was investigated at different concentrations of GSH (Figure 2A). In the medium free of GSH, the release rate of ICG was 36.22% within 120 h, whereas in the medium with a high concentration of GSH, the released ICG reached 74.50%. This result indicated that, with increasing GSH concentration, the cleavage speed of the disulfide bond in HD/CNTs-SS-ICG was accelerated, leading to the rapid release of ICG from the nanosystem. The release of DOX from HD/CNTs-SS-ICG@DOX was evaluated under different temperatures and pH conditions. This nanosystem exhibited a dual-temperature and pH-sensitive drug release property. In the medium simulating normal conditions (at 25 °C and pH 7.4), the release rate of DOX was only 29.41% after 120 h, while under the conditions with a high temperature and low pH (at 45 °C and pH 5.0), the release rate was up to 94.25% (Figure 2B). These results imply that DOX release could be enhanced by photothermal effects on the local tumor environment. Furthermore, the DOX release profiles of various nanosystems are shown in Figure 2C,D. The Higuchi model was used to fit the drug release curve. It was found that the release rate constants (K) of SWCNTs@DOX, CNTs@DOX, CNTs-SS@DOX, and HD/CNTs-SS@DOX were 0.1644, 0.6606, 2.5381, and 5.6345, respectively, at pH 7.4. The K of these nanosystems was 0.9711, 1.6665, 1.8658, and 7.2762 at pH 5, respectively. All the nanosystems exhibited a more rapid drug release at pH 5.0 than that at pH 7.4, and the functionalized systems had a higher release amount than unmodified samples within 120 h.

### 2.3. Photothermal Performance Verification

Photothermal analysis was performed at an excitation wavelength of 808 nm due to its advantages of efficient tissue penetration, minimal absorption by water molecules, cells, and tissue components, as well as its biosafety [22,23]. The photothermal conversion property of ICG was verified under NIR irradiation of 5 W/cm^2^; the result indicated that the heating rate increased with increasing ICG concentration (Figure 3A). After 210–240 s of irradiation, the high temperature induced by free ICG with different concentrations rapidly decreased, owing to the photolysis property. Thus, it was crucial to load ICG by nanocarriers to improve its photothermal stability. The HD/CNTs-SS-ICG solution exhibited an excellent response to an irradiation intensity of 5 W/cm^2^ (Figure 3B,C), and the temperature increased to 45 °C within 30 s after NIR irradiation at this intensity. The tumor cells could undergo thermal ablation at this temperature, inducing further necrosis.

The photothermal properties of HD/CNTs-SS and HD/CNTs-SS-ICG@DOX were also verified under NIR irradiation of 5 W/cm^2^ (Figure 3D). HD/CNTs-SS had an inferior photothermal conversion performance, which only increased the environmental temperature to 35.2 °C at 300 s. Conversely, HD/CNTs-SS-ICG@DOX could rapidly increase the temperature to above 40 °C, and the final temperature reached approximately 43.5 °C. These results demonstrated that it was indispensable to incorporate ICG to manufacture a nanosystem to trigger PPT effects, and that the loading of DOX almost did not affect the photothermal conversion performance of HD/CNTs-SS-ICG. Furthermore, HD/CNTs-SS-ICG solutions maintained a good photothermal performance in four heating–cooling cycles, with photothermal conversion efficiencies greater than 60% (η1 = 63.89%, η2 = 73.05%, η3 = 85.51%, η4 = 90.64%). The incremental η values could result from the accumulatively released ICG with increasing cycles (Figure 3E).

### 2.4. Cell Viability Assessment

Herein, the cytotoxicity of various SWCNTs against MCF-7 cells was assessed within 24 h (Figure 4A). Free ICG showed higher cytotoxicity than SWCNT nanoparticles. At concentrations of 50 μg/mL or 100 μg/mL, functionalized HD/CNTs-SS-ICG had a higher cell survival rate compared with other SWCNTs (above 85%). The hemolysis rate results shown in Figure 4C,E indicated that the hemolysis rates of various SWCNTs were all less than 5% at 10–100 μg/mL, showing good biological safety.

In the HD/CNTs-SS-ICG@DOX treatment group, after 5 W/cm^2^ of NIR irradiation for 5 min, the cell survival rate decreased to 42.66%. The cell lethal rate was nearly 3.61% and 13.55% higher than that of HD/CNTs-SS@DOX and HD/CNTs-SS-ICG+NIR treatment, respectively, verifying the superior synergistic tumor inhibitory effect over chemotherapy or PTT alone (Figure 4D). The survival rate of IPEC-1 cells by free DOX treatment was approximately 55.17%, suggesting that chemotherapy drugs had an inhibitory effect on normal cell growth. By the same treatment with HD/CNTs-SS@DOX, the survival rate of IPEC-1 cells was much higher compared to that of MCF-7 cells (by approximately 40%), which was likely due to the enhanced targeting effect of nanocarriers originating from HD modification towards tumor cells. Similarly, in the HD/CNTs-SS-ICG@DOX+NIR treatment group, the viability of IPEC-1 cells was nearly 30% higher than that of MCF-7 cells, indicating that synergistic chemotherapy and PTT had higher biological safety in normal cells (Figure 4B).

### 2.5. Cellular Uptake and Intracellular Distribution

The cellular fluorescence of ICG and DOX was observed using a fluorescence microscope to analyze the uptake of the nanosystem by MCF-7 cells. As shown in Figure 5A–C, only a weak fluorescence signal was detected in cells treated with free DOX or ICG, owing to the fact that free drug molecules were transferred into cells only by passive mobility or diffusion. Conversely, as the nanocarriers, functionalized SWCNTs were apt to be internalized by cells via direct penetration or endocytosis. Thus, various nano-samples exhibited a more efficient cellular uptake than free drugs, and the fluorescence intensity was increased gradually with the improvement in surface modification for nanocarriers. Furthermore, FCM was used to quantitatively assess the intracellular fluorescence intensity. As shown in Figure 5D and Appendix A, various nanosystems could be effectively internalized by cells in a time-dependent manner, indicating the gradual accumulation of delivery carriers in tumor cells. HD/CNTs-SS-ICG@DOX showed a higher uptake efficiency compared with other formulations; the fluorescence intensity at 24 h was almost 4.0-fold that of free DOX. This result should be attributed to the fact that surface modification would enhance the water dispersion of SWCNTs and improve their targeting affinity to tumor cells, being conducive to efficient delivery and subsequent intracellular drug release.

The intracellular distribution of FITC-labeled HD/CNTs-SS nanoparticles in MCF-7 cells was investigated by confocal microscopy. As shown in Appendix A, most of the green fluorescence of FITC colocalized with the red fluorescence of Lyso-tracker after treatment with HD/CNTs-SS@FITC for 4 h. After 16 h of co-incubation, it could be observed that HD/CNTs-SS@FITC gradually translocated from the lysosome to the cytoplasm, and a small number of nanoparticles accumulated into the nucleus. At 24 h, FITC fluorescence overlapped with Hochest blue fluorescence, becoming higher; meanwhile, the intensity of red fluorescence was significantly reduced, implying that the nano-samples achieved lysosomal escape and could locate the nucleus to exert an antitumor effect.

### 2.6. ROS Measurement

The effects of various formulations on ROS levels in MCF-7 cells were assessed and the results are shown in Figure 5E–G. The 2′,7′-dichlorodihydrofluorescein (DCF) fluorescence of free DOX was relatively weak when compared to CNT nano-formulations, and the red fluorescence from DOX was slightly enhanced under NIR irradiation, indicating that irradiation was helpful for the cellular uptake of drugs [24]. After treatment with HD/CNTs-SS@DOX and HD/CNTs-SS-ICG+NIR for 24 h, the green fluorescence of DCF increased. However, after 24 h of treatment with HD/CNTs-SS-ICG@DOX+NIR, both the red and green fluorescence reached their highest intensity. This result suggested that synergistic therapy can effectively increase DOX uptake and ROS production, leading to triggering cell apoptosis and the necrosis pathway. The enhanced intracellular fluorescence of DCF with the extension of the treatment time indicated that the synergistic antitumor effect of chemotherapy and PTT was also time-dependent.

### 2.7. Cell Apoptosis and Cell Cycle Variation Analysis

The apoptosis rate in MCF-7 cells treated with various formulations was investigated by FACS (Figure 6A–D). After 16 h of treatment, the apoptosis rate of cells treated with free DOX was 2.41%, while that of the HD/CNTs-SS@DOX treatment group increased to 2.64% and that of the HD/CNTs-SS-ICG+NIR group further increased by nearly 18%. More notably, the apoptosis rate of the HD/CNTs-SS-ICG@DOX+NIR treatment reached 44.1%, and the total cell death rate (apoptosis rate + necrosis rate) was 48.62%, indicating that high-intensity light irradiation led to necrosis and the rapid ablation of tumor cells. Similarly, the apoptosis rate in the HD/CNTs-SS@DOX and HD/CNTs-SS-ICG+NIR groups after 48 h of treatment was 28.37% and 36.42%, respectively, whereas the total death rate caused by the HD/CNTs-SS-ICG@DOX+NIR treatment was up to 70.2%. It could be concluded that synergistic therapy mediated by HD/CNTs-SS-ICG@DOX systems had a better antitumor effect than chemotherapy or PTT alone to achieve a 1 + 1 > 2 efficacy.

FACS was used to investigate the cell cycle distribution of MCF-7 cells after different treatments (Figure 7A–D). Compared with the control group, the S-phase fraction (SPF) of cells treated with HD/CNTs-SS@DOX for 16 h and 48 h increased by 40% and 47%, respectively, indicating that loading DOX by CNT carriers could affect the cycle distribution of MCF-7 cells significantly. Furthermore, the SPF of the HD/CNTs-SS-ICG+NIR group increased by 10%. In particular, the SPF and G2 phase cells after HD/CNTs-SS-ICG@DOX+NIR treatment increased more markedly, which implied that synergistic treatment with PTT and chemotherapy increased the number of MCF-7 cells in the S phase and arrested the cell cycle in the G2 phase. Therefore, the number of cells in the M phase was reduced to achieve an antitumor effect by interfering with cell division and decreasing DNA synthesis.

### 2.8. Evaluation of the Antitumor Efficacy In Vivo

Model animals were established using MCF-7 cells to explore the photothermal properties of the nanosystems for in vivo therapy (Figure 8A,B). Female BALB/c nude mice were randomly divided into five groups: blank control (NS), free DOX, HD/CNTs-SS@DOX, HD/CNTs-SS-ICG+NIR, and HD/CNTs-SS-ICG@DOX+NIR. The photothermal effects of each sample after exposure to NIR light for 5 min were then evaluated. The temperature of the HD/CNTs-SS-ICG and HD/CNTs-SS-ICG@DOX groups rose obviously within the first 1 min of NIR irradiation, reaching 48 °C and 45 °C, respectively. Then, the temperature rose slowly and stabilized at about 50 °C after 2 min. In the free ICG group, the temperature was stable at 40 °C after NIR irradiation, indicating that ICG combined with CNTs could synergistically improve their photothermal conversion capacity. There was no significant temperature variation at the tumor site in the NS treatment group, and treatment with HD/CNTs-SS-ICG and HD/CNTs-SS-ICG@DOX rapidly led to a temperature rise at the tumor site, reaching the optimal temperature for triggering the photothermal effect.

The procedure for the in vivo assessment of antitumor therapy is shown in Figure 9A. The tumor volume of the mice was measured and photographed every 3 days (Figure 9B,G). In the chemotherapy group (free DOX), the tumor volume decreased in the first six days, and then increased to some extent, which may be due to the increased drug resistance of the tumor after repeated administration. The tumor volume of nude mice in the chemotherapy group (HD/CNTs-SS-ICG@DOX) and PTT group (HD/CNTs-SS-ICG+NIR) did not change significantly, which further confirmed that both chemotherapy and PTT could inhibit tumor growth. On the contrary, in the HD/CNTs-SS-ICG@DOX+NIR group, the tumor volume continued to decrease within 15 days, which proved that synergistic therapy could stably inhibit tumor growth and had a good antitumor effect. At the endpoint of the experiments, the tumor was resected and weighed to evaluate the in vivo antitumor effect by different treatments. As shown in Figure 9D–F, simultaneous chemotherapy and PTT mediated by the HD/CNTs-SS-ICG@DOX system had a more potent inhibitory activity against tumor growth, as tumor eradication in one mouse and a dramatically decreased tumor size in other four mice were observed after 15 days of treatment. Furthermore, the tumor weight of the HD/CNTs-SS-ICG@DOX+NIR group was less and the TGIR was much higher than other treatments (up to 83.77%, Figure 9E). These results confirmed that synergistic therapy with chemotherapy and PTT could stably inhibit tumor growth and achieve a better therapeutic effect.

The body weight changes in nude mice during the entire treatment were monitored to further assess the biosafety of various nanosystems (Figure 9C). With the extension of treatment time, the body weight in each treatment group increased slightly, suggesting that the different treatment schemes did not lead to significant systemic toxicity. Compared to the HE staining of normal organs and NIR-irradiated organs in the blank control (Appendix A), no intense damage to normal organs and tumor tissues was observed after NIR irradiation, indicating that PTT had little effect on normal tissues and thus verifying the biosafety of NIR irradiation used in PTT. Finally, free DOX and various nano-drug treatments inflicted no remarkable organ damage on nude mice; only the synergistic treatment group exhibited a marked necrosis effect on tumor tissues, which favored achieving a better antitumor effect (Figure 8C).

## 3. Materials and Methods

### 3.1. Chemicals and Reagents

Dithiodipropionic acid (DTDP), 4-dimethylaminopyridine (DMAP), triethylamine (TEA), tribromoethanol, and tert-amyl-alcohol were obtained from McLean Reagent Company (Shanghai, China). NH_2_-PEG-OH (MW2000) was purchased from the Xi’an Ruixi Biotechnology Co., Ltd (Xi’an, China). HA (MW3000) was obtained from Yuanye Biotechnology Co., Ltd. (Shanghai, China). Doxorubicin (DOX) was purchased from Dalian Meilun Biotechnology Co., Ltd. (Dalian, China). ICG-NH_2_ was obtained from Xi’an Qiyue Biotechnology Co., Ltd. (Xi’an, China). SWCNTs were obtained from the Chengdu Institute of Organic Sciences, Chinese Academy of Sciences. *N*,*N*-dimethylformamide (DMF), GSH, *N*-hydroxythiosuccinimide (NHS), and paraffin were obtained from Aladdin Reagent Co., Ltd. (Shanghai, China). Cyclohexylcarbodiimide (DCC) was obtained from Ron Technology Co., Ltd. (Beijing, China). Acetyl chloride and ice ether were purchased from Damao Chemical Reagent Co., Ltd. (Tianjin, China). Dimethyl sulfoxide (DMSO), xylene, absolute ethanol, and glacial acetic acid were purchased from Fuyu Chemical Reagent Co., Ltd. (Tianjin, China). HCl was purchased from Komeo Chemical Reagent Co., Ltd. (Tianjin, China). Carbodiimide (EDAC) was acquired from Shanghai Haohong Biomedical Technology Co., Ltd. (Shanghai, China). MCF-7 human breast cancer cells and IPEC-1 cells were purchased from Shanghai Bogu Biotechnology Co., Ltd. (Shanghai, China). RPMI 1640, PBS, trypsin, and Hoechst33342 were purchased from Solebo Biotechnology Co., Ltd. (Guangzhou, China). Fetal bovine serum (FBS) was purchased from Zhejiang Tianhang Biotechnology Co., Ltd. (Hangzhou, China). MTT was purchased from Bioengineering (Shanghai) Co., Ltd. (Shanghai, China). Cell cycle and apoptosis detection kit, AnnexinV-FITC apoptosis detection kit, ROS detection kit, and lysosomal red fluorescent probe (Lyo-Tracker Red) were purchased from Biyuntian Biotechnology Co., Ltd. (Shanghai, China). Triton Xmure 100 (TritonX-100) was purchased from Sigma Aldrich Trading Co., Ltd. (Shanghai, China). Paraformaldehyde 4% universal tissue fixative was purchased from Guangzhou Shuopu Biotechnology Co., Ltd. (Guangzhou, China). Normal saline (NS) was purchased from Henan Kelun Pharmaceutical Co., Ltd. (Anyang, China). Hematoxylin and eosin dye was purchased from Hunan Yunbang Pharmaceutical Co., Ltd. (Changsha, China).

### 3.2. Cell Culture

In this study, MCF-7 breast cancer cells obtained from American Type Culture Collection (ATCC, Manassas, VA, USA) were used as the model. The cells were cultured in RPMI-1640 medium supplemented with 9% FBS and incubated under the following culture conditions: 5% CO_2_, 90% humidity, and a constant temperature of 37 °C. IPEC-1 porcine small intestine epithelial cells were used as the control model.

### 3.3. Experimental Animals

All animal experiments followed the guidelines of the National Institutes of Health and the rules and guidelines approved by the Ethics Committee of the Animal Center of Zhengzhou University. BALB/c female mice aged 4–6 weeks (body weight: 14–15 g) were purchased from Beijing WeitongLihua Experimental Animal Technology Co., Ltd. MCF-7 cells (1 × 10^7^) and 0.2 mL of NS were subcutaneously injected (s.c.) into the right forelimb of mice to establish the tumor-bearing nude mouse model [25].

### 3.4. Development of the Functionalized SWCNT Delivery System

#### 3.4.1. Synthesis of Nano-Sized CNTs

First, 50 mg of SWCNTs was added into nitric acid, then the sample was ultrasonicated (200 W, 53 kHz) for 3 h and left to rest overnight. Subsequently, it was heated at 90 °C for 24 h and the supernatant was removed by centrifugation at 24,000 rpm for 30 min. The precipitate was washed to neutral with ultrapure water, and CNTs were obtained after vacuum-drying.

#### 3.4.2. Preparation of CNTs-SS-ICG

**Synthesis of DTDPA**: The 1 g of DTDP was added into 5 mL of acetyl chloride in an oil bath at 65 °C for 5.5 h. The sample was rotary-evaporated at 50 °C, and white flocculent appeared after 20 mL of glacial ether was added. Afterward, the suspension was suction-filtered and washed with glacial ether. The residue was collected and subjected to vacuum-drying at 30 °C, and the white precipitate obtained was DTDPA.

**Preparation of NH_2_-PEG-SS-COOH:** DTDPA, NH_2_-PEG-OH, DMAP, and TEA were dissolved in DMF. The above mixture was reacted at 600 rpm at either 37 °C or 45 °C for 24 h. Subsequently, ice ether (50 mL) was added into the mixture and further subjected to centrifugation at 1000 rpm for 8 min and then maintained at 4 °C; the procedure was repeated twice. The white precipitate, which was NH_2_-PEG-SS-COOH, was collected and dried under a vacuum at 25 °C.

**Preparation of CNTs-SS:** The 30 mg of CNTs was dispersed in DMSO (9 mL) and further ultrasonicated (200 W, 53 kHz) with an ultrasonic cleaner for 3 h. The 1 mL of NHS and 1 mL of DCC (dissolved in DMSO) were introduced to a CNT suspension to activate carboxyl groups. After 12 h, 1 mL of NH_2_-PEG-SS-COOH (31 mg/mL in DMSO) and 1 mL of DMAP (150 mg/mL in DMSO) were added to the activated CNT solution. After reacting at 600 rpm and 37 °C for 24 h, the final mixture was dialyzed and freeze-dried to obtain CNTs-SS.

**Preparation of CNTs-SS-ICG:** The 20 mg of CNTs-SS was dispersed in DMSO (9 mL) and then activated by 50 mg of NHS and 40 mg of DCC. The ICG-NH_2_ (1 mg/mL in DMSO) and DMAP (1 mL; 25 mg/mL DMSO solution) were catalyzed and allowed to react with CNTs-SS in the absence of light for 24 h. After that, the mixture was centrifuged at 12,000 rpm for 20 min and the supernatant was collected to determine the loading efficiency of ICG by high-performance liquid chromatography (HPLC). CNTs-SS-ICG were obtained after the precipitation was lyophilized.

#### 3.4.3. Preparation of HD/CNTs-SS-ICG@DOX

**Synthesis of HA-DMPE:** The 20 mg of HA and 11.4 μL of EDAC were dissolved in ultrapure water (5 mL) and stirred at 37 °C for 2 h. Subsequently, HA was activated by adjusting the pH to 8.4 with 0.1 M borate buffer. The 50 mg of DMPE was dissolved in ether (5 mL) and further rotary-evaporated to form the lipid membrane. The activated HA solution was poured into the lipid membrane. After ultrasonication (200 W, 53 kHz) for 30 min, the mixture was further reacted at 37 °C for 24 h. The final mixture was dialyzed and freeze-dried to obtain HA-DMPE (HD).

**Preparation of HD/CNTs-SS-ICG@DOX:** The 10 mg of CNTs-SS-ICG was dispersed in PBS (20 mL). After 2 h of ultrasonication (200 W, 53 kHz), DOX solution (1 mg/mL) was added and allowed to react at 37 °C for 4 h. The resulting products (CNTs-SS-ICG@DOX) were centrifuged at 12,000 rpm for 20 min. The precipitation was collected and mixed with HD solution (1 mg/mL in PBS). After ultrasonic dispersion, the mixture was reacted at 37 °C for 2 h and further centrifugated at 12,000 rpm for 20 min. The amount of free DOX in the supernatant was determined and the precipitation was collected and lyophilized to obtain HD/CNTs-SS-ICG@DOX.

### 3.5. Characterization

The particle size and zeta potential of SWCNTs, CNTs, CNTs-SS, CNTs-SS-ICG, HD/CNTs-SS-ICG, and HD/CNTs-SS-ICG@DOX were analyzed using a Nano-ZS90 mode laser particle size analyzer and the Nano-ZS90 laser potentiometer (Malvern Instruments, Malvern, UK) [26]. The morphology and microstructure of the samples (1 mg/mL in ultrapure water) were observed using a HITACHI HT7700 transmission electron microscope (Hitachi High-Tech (Shanghai) Co., Ltd, Shanghai, China) and a HITACHI SU8020 scanning electron microscope (Hitachi High-Tech (Shanghai) Co., Ltd, Shanghai, China), respectively [27].

The 5 mg of dried DTDP, DTDPA, NH_2_-PEG-OH, NH_2_-PEG-SS-COOH, CNTs, and CNTs-SS powders were, respectively, weighed for Fourier transform infrared (FT-IR) detection to verify the formation of characteristic functional groups [28]. Aqueous solutions (0.1 mg/mL) of ICG, DOX, CNTs-SS, CNTs-SS-ICG, HD/CNTs-SS, HD/CNTs-SS@DOX, and HD/CNTs-SS-ICG@DOX were scanned with a MAPADA ultraviolet (UV-vis) spectrophotometer at a wavelength of 280–800 nm. The 0.5 mL of DTDP, DTDPA, NH_2_-PEG-OH, and NH_2_-PEG-SS-COOH were dissolved in DMSO-d6 (+0.03%TMS) and transferred into a nuclear magnetic tube for the measurement of ^1^H NMR spectra using a 600 MHz nuclear magnetic resonance spectrometer.

Powder samples (0.2 g) of SWCNT, CNT, CNT-SS, and HD/CNT-SS were weighed and tested with a BRUCKER D8 ADVANCE X-ray diffractometer (Bruker AXS, Berlin, Germany). The scanning range was set between 5 and 80°, with a scanning step of 0.02° and scanning speed of 5°/min. Various samples were uniformly laid on glass slides, pressed, and smoothed. Raman scattering was measured using a Renishaw on a Via Raman spectrometer at an excitation wavelength of 633 nm in the range of 100–3500 cm^−1^. Two flat stainless-steel modules were pressed and measured on the sample table of a Thermo escalab250XiX X-ray photoelectron spectrometer (Thermo Fisher Scientific, Waltham, MA, USA) with double-sided adhesive. The narrow scan elements were carbon (C), oxygen (O), nitrogen (N), and sulfur (S), which were used to determine the types of elements present on the sample surface.

### 3.6. Photothermal Effect Analysis

The temperature curves of the PBS and HD/CNTs-SS-ICG were drawn under 808 nm NIR irradiations of 4 W/cm^2^, 5 W/cm^2^, and 6 W/cm^2^ for 5 min, respectively. Temperature changes in the four cycles were recorded, and the photothermal conversion efficiency was obtained from the heat transfer curves [29]. The photothermal conversion efficiency was calculated according to the following formula: η = hA(△Tmax − △Tmax.PBS)/I(1 − 10^−Aλ^),
where h = 0.59 W/(m^2^.K), A is the surface area of the container (4.5 × 10^−4^ m^2^), Tmax is the temperature change in the sample solution at the highest steady temperature, Tmax.PBS is the temperature change in PBS at the highest steady temperature, I is the laser power, and Aλ is the absorbance of the sample solution at 808 nm, respectively [30,31].

### 3.7. Drug Loading and Encapsulation Efficiency

Following the synthesis of CNTs-SS-ICG and HD/CNTs-SS-ICG, the supernatant was collected to determine ICG using HPLC. The load efficiency of ICG was calculated according to the following formula: load efficiency (%) = (Total _ICG_ − Free _ICG_)/Total _ICG_ × 100%.

SWCNTs, CNTs, CNTs-SS, CNTs-SS-ICG, and CNTs-SS-ICG (10 mg) were suspended in 20 mL of PBS. After ultrasonication for 2 h, 2 mg of DOX solution at a concentration of 1 mg/mL was added and ultrasonicated for another 2 h [32]. The mixture was centrifuged at 12,000 rpm for 20 min and the supernatant was taken to measure the amount of free DOX using UV-vis at 480 nm. DOX encapsulation efficiency and drug loading were calculated according to the following formulas: encapsulation rate (%) = (Total _DOX_ − Free _DOX_)/Total _DOX_ × 100%, drug loading (%) = (Total _DOX_ − Free _DOX_)/(Total amount of carrier + Total _DOX_) × 100%.

### 3.8. In Vitro Drug Release

The HD/CNTs-SS-ICG was dispersed in PBS containing 0, 2, 5, or 10 mM GSH and oscillated at 160 rpm and room temperature. ICG concentration was determined by HPLC at preset time intervals.

The HD/CNTs-SS-ICG@DOX was dispersed in 5 mL of PBS and then suspended in 50 mL of PBS at different temperatures and pH, respectively. Furthermore, SWCNTs@DOX, CNTs@DOX, CNTs-SS@DOX, CNTs-SS@DOX, and HD/CNTs-SS-ICG@DOX were suspended in PBS at pH 5.0 or 7.4, respectively. All the samples were oscillated at 160 rpm and at preset time intervals, 2 mL of supernatant was taken out, and the percentage of released DOX was determined by UV-vis.

### 3.9. Cytotoxicity and Hemolysis of Nanoparticles

The cytotoxicity of SWCNTs, CNTs, CNTs-SS, ICG, CNTs-SS-ICG, and HD/CNTs-SS-ICG towards MCF-7 cells was assessed by MTT assay [33]. The cells were cultured in 96-well plates overnight; then, the culture medium was replaced by fresh medium containing different nanoparticles. After 24 h, 20 μL of MTT solution was added for further incubation for 4 h. Subsequently, the medium was removed and 100 μL of DMSO was added to disperse the formazan precipitate, and the multi-functional plate reader (Varioskan Flash, ThermoFisher Scientific, Waltham, MA, USA) was used to measure the optical density of the solution at 490 nm [34,35]. Hemolysis of various nanoparticles was evaluated using 1% Triton and NS as positive and negative control, respectively. The hemolysis rate was calculated using the following formula: hemolysis rate = (Experimental group A540 − Negative control A540)/(Positive control A540 − Negative control A540) × 100%.

### 3.10. In Vitro Antitumor Activity

The inhibitory effect of DOX, HD/CNTs-SS@DOX, ICG (2.0 μg/mL), HD/CNTs-SS-ICG, and HD/CNTs-SS-ICG@DOX against MCF-7 cells and IPEC-1 cells was also evaluated by MTT assay. Lastly, the absorbance of each well was measured at 490 nm to calculate cell viability, as mentioned above. The cells incubated with nano-formulations were exposed to NIR laser irradiation (5 W/cm^2^, 808 nm) for 5 min, and the cells were incubated for another 4 h to investigate the photothermal influence on cytotoxicity.

### 3.11. Cellular Uptake

The cellular uptake of various nano-formulations was determined by fluorescence microscope and flow cytometry (FCM). The MCF-7 cells were cultivated overnight at a density of 2 × 10^5^ cells per well in 12-well plates. After incubating each sample for different times (4 h or 24 h), the cells were washed twice with PBS and centrifuged at 1000 rpm for 5 min to collect cells. Finally, the cells were resuspended in PBS, and the average fluorescence intensity of DOX and ICG was quantified using a flow cytometer (Thermo Fisher Scientific, Waltham, MA, USA) and the cellular uptake was observed by a fluorescence microscope [36].

### 3.12. Intracellular Reactive Oxygen Species (ROS) Assessment

The MCF-7 cells were seeded in 12-well plates overnight at a density of 2 × 10^5^ cells per well at 37 °C. Then, the cells were treated with free DOX, HD/CNTs-SS@DOX, free ICG, HD/CNTs-SS-ICG, HD/CNTs-SS-ICG+NIR, and HD/CNTs-SS-ICG@DOX+ NIR, respectively. After 4 h or 24 h of incubation, the cells in the last two treatment groups were irradiated with an 808 nm NIR laser at 5 W/cm^2^ for 5 min. Next, the serum-free medium containing 10 μM of DCFH-DA was added into each well. The cells were incubated for another 20 min in darkness and then washed with PBS and trypsinized. Lastly, the cell suspension was used to detect intracellular ROS levels using Fluorescence-activated Cell Sorting (FACS) and a fluorescence microscope, respectively.

### 3.13. Cell Apoptosis and Cell Cycle Distribution Detection

The MCF-7 cells were treated with NS, free DOX, HD/CNTs-SS@DOX, ICG+NIR, HD/CNTs-SS-ICG+NIR, and HD/CNTs-SS-ICG@DOX+NIR, respectively. After co-incubation for 16 h or 48 h, the cells in the last three treatment groups were irradiated with an 808 nm NIR laser. Next, the cells were incubated for another 4 h and then washed with PBS, trypsinized, and centrifugated at 1000 rpm. Afterward, 500 μL of binding buffer containing PI/Annexin V staining solution was added for further incubation. The cell apoptosis ratio was determined by FACS. The cells in NS treatment group were used as the control group. The experimental procedure of cell cycle measurement was similar to the above process. The collected cells were resuspended in cold anhydrous ethanol overnight to avoid structural damage to the cytomembrane [37]. The cell suspensions were subsequently centrifugated to remove ethanol, and 500 μL of PI buffer solution was added for incubation at 37 °C in darkness. The cells in yjr NS treatment group were used as the control group to determine the cell cycle variation.

### 3.14. In Vivo Antitumor Effect Evaluation

The MCF-7 cell suspension was injected subcutaneously into the mice. When the tumor size reached 50–80 mm^3^, the mice were randomly divided into five groups. Mice in each group were subcutaneously injected with 50 μL of NS, DOX (3 mg/kg), HD/CNTs-SS@DOX (15 mg/kg), HD/CNTs-SS-ICG (15 mg/kg), or HD/CNTs-SS-ICG@DOX (15 mg/kg). The latter two groups of mice were anesthetized with a mixture of tribromoethanol and tert-*amyl* alcohol and their tumor sites were exposed to an 808 nm laser (5 W/cm^2^, 5 min). Temperature variations at the tumor site were recorded and photographed by infrared thermal imaging. After continuous treatment for 15 days, the antitumor effect on each group of mice was evaluated by monitoring tumor volume and body weight every 3 days [38]. The tumor volume was calculated according to the following formula: tumor volume= (length × width^2^)/2.

Lastly, the mice were sacrificed to remove the tumor and major organs, and the resected tumor was weighed and photographed. The tumor growth inhibition rate (TGIR) was calculated according to the following formula: TGIR = [(W_c_ − W_t_)/W_c_] × 100%,
where W_c_ and W_t_ represent the average tumor weight of the control group and treatment group, respectively [39]. The major organs and tumor tissues after perfusion were fixed overnight in 4% paraformaldehyde. The tissues were soaked in a mixture of xylene and molten paraffin (volume ratio 1:1) and soaked overnight. The organized wax blocks were sliced with a slicer and affixed to the slide. After dewaxing, the wax blocks were stained with hematoxylin and eosin (HE).

### 3.15. Statistical Analysis

All data are prepared as mean ± SD and the mean values were considered significantly different when *p** < 0.05, *p*** < 0.01, or *p**** < 0.001. ANOVA was used for the statistical analysis of all the treatment groups by SPSS software (version 9.1).

## 4. Conclusions

In this study, DTDP was used as an S-S donor to manufacture a TME-responsive nano-delivery system for antitumor therapy through mediating synergistic chemotherapy and PTT. The HD/CNTs-SS-ICG@DOX system exhibited a powerful antitumor effect both in vitro and in vivo under NIR irradiation. In TME, a high concentration of GSH would break S-S in CNT carriers, and the S-S fracture is reduced to -SH, causing the release of ICG. At the same time, NIR light is applied externally to convert the light energy into heat energy. The temperature increase caused by PTT further promoted drug release and achieved efficient coordination between chemotherapy and PTT. Furthermore, the HD/CNTs-SS-ICG@DOX system had a low cytotoxicity and hemolysis rate, and the cell mortality rate of synergistic therapy was notably higher than that of chemotherapy or PTT used alone. In tumor-bearing nude mice, the HD/CNTs-SS-ICG@DOX system also showed good biosafety and could mediate synergistic chemotherapy and PTT to achieve enhanced antitumor efficacy. This study, which fully exploits the advantages of each therapy, has shown promising results in both cellular and mouse models. However, the limitation of light penetration to deep tumors remains the biggest challenge, and more clinic-based in vivo antitumor studies are needed to explore more effective strategies to design and develop PTAs with better penetration and nanocarriers with higher safety. In conclusion, the “intelligent” drug delivery system developed in this study has tumor-targeted delivery, a microenvironmental response to drug release, and excellent photothermal properties. The nanosystem developed has extensive prospects in antitumor treatment applications as an efficient collaborative platform for chemotherapy and PTT.

## Data Availability

Data are contained within the article.

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
