# Peer review of "Synergistic Photothermal Therapy and Chemotherapy Enabled by Tumor Microenvironment-Responsive Targeted SWCNT Delivery"

_ijms, 2024, doi:10.3390/ijms25179177_

Round 1

Reviewer 1 Report

Comments and Suggestions for Authors

Here, Yang et al. developed a novel tumor-targeted nanosystem combining photothermal therapy (PTT) and chemotherapy, demonstrating enhanced antitumor efficacy and biosafety in both in vitro and in vivo models. The study highlights the potential of their system for effective cancer treatment by leveraging tumor microenvironment-responsive drug release. The study seems to be promising with extensive studies. I have some minor queris:

1) The manuscript should clearly highlight the novel aspects of the research and how it advances the current state of knowledge in the field of photothermal therapy and chemotherapy.

2) The methods section needs more detailed descriptions, especially regarding the synthesis and characterization of the various SWCNT nanocarriers.

3) The in vivo antitumor efficacy evaluation could benefit from additional data points and extended observation periods. Including more detailed analyses of the tumor growth curves, survival rates, and potential side effects over a longer duration will provide a more thorough assessment of the therapeutic potential and safety of the developed nanosystem.

4) A discussion on the potential limitations and challenges of the proposed synergistic therapy system would be beneficial. Addressing issues such as the stability of the nanocarriers in biological environments, potential toxicity concerns, and the scalability of the synthesis process will provide a balanced view of the study's findings.

5) The manuscript would benefit from a more in-depth mechanistic explanation of how the synergistic effect between photothermal therapy and chemotherapy is achieved. Including discussions on the cellular uptake pathways, intracellular distribution, and specific interactions between the nanocarriers and tumor cells will add depth to the understanding of the therapeutic mechanism.

6) The Scheme 1 is very good, but lacks information in the figure caption. Add some more details. Also, increase the font size. 

Comments on the Quality of English Language

Minor editing of English language required.

Reviewer 2 Report

Comments and Suggestions for Authors

The authors of the study synthesized multifunctional nanoparticles for dual photothermal and chemotherapy applications. The hypothesis and rationale are well-founded and supported by various experiments. However, there are a few discrepancies in the manuscript that need to be addressed before it can be accepted for publication. Below, I have listed some major and minor comments for your consideration.

1)      Lines 49-50: The statement, “Combining PTT and chemotherapy can reduce the required dose of chemotherapeutic drugs, reduce the expression of heat shock proteins in tumor cells, improve the curative effect, and reduce adverse reactions,” is accurate but lacks clarity.

a)      While combining chemotherapy with other adjuvant therapies may reduce the required drug dose, how does the reduction in heat shock proteins benefit the drug’s cytotoxic mechanism?

2)      Lines 51-53: “PTT generates enough heat energy to stimulate the release of chemotherapy drugs, increase their distribution in tumor cells, and achieve an efficient combination with chemotherapy.”

a)      The dual combination mechanism needs further discussion, as the mechanism of action is not clearly explained.

3)      Line 60: The term "low dark toxicity" may not be familiar to many researchers outside the field. Please provide a reference. Additionally, ICG is FDA-approved for imaging; this necessitates clarity or evidence to state.

4)      Scheme 1: The DOX color is not clearly visible in the synthesis of the final product. Clarify whether the dosing is subcutaneous or other route and if the particles cross the blood-brain barrier (BBB), considering the final product size. The figure legend could be rephrased accordingly, and it would be good practice to mention how these figures were sketched.

5)      Figures 1-8: The clarity of these figures is too poor. Please insert high-resolution images.

6)      Table 2: The method used for analysis is missing from the title.

7)      Figure 2A: The Y-axis is labeled "accumulative release." It is recommended to use "cumulative release" to be consistent with Figures 2B, C, and D. Additionally, there are typos in all the figures.

8)      Line 233: The potential PTT use of NIR 808 nm light is usually <2 W/cm². You have used 5 W/cm². How is this safe?

9)      Figure 3 Legends: Mention the concentration of ICG used (free ICG and ICG conjugated CNTs [HD/CNTs-SS-ICG]).

10)  Line 343: There is a typo: “(apoptosis rate + necrosis rates).”

11)  Line 453: Is the FBS concentration 9%?

12)  Particle Synthesis Process: The conditions of ultrasonication used should be mentioned, including the type of sonicator.

13)  Section 3.10 In vitro antitumor activity test: What is the concentration of ICG used (free ICG and ICG conjugated CNTs [HD/CNTs-SS-ICG])? What was the temperature rise when induced with NIR laser for 5 minutes at 1 W/cm², and why was this intensity chosen? There is a discrepancy in the temperature sensitivity tests, as 3, 4, and 5 W/cm² were used, and the ICG concentration (free or within the conjugated nanoparticles) is not mentioned.

14)  Section 3.12 Intracellular reactive oxygen species (ROS) assessment: Why was the NIR laser intensity 5 W/cm² and not 1 W/cm²? What was the temperature change during the irradiation window? Are the concentrations of free ICG or within the conjugated nanoparticles the same?

15)  Section 3.14 In vivo antitumor effect evaluation: What was the route of administration? What was the NIR-induced intensity and time? What was the temperature change during the induction in mice? This information is missing.

Critique of the Study:

The study used a NIR laser at 5 W/cm². Several publications suggest minimizing adverse effects, including over-burning the tumor site (as depicted in Figure 9g) and other adverse effects, by using NIR up to 1-2 W/cm². This point is clearly missing in the in-vitro and in-vivo experiments. Additionally, the authors claim in the introduction that CNTs are photothermal agents (PTAs): “Nanomaterials commonly used as delivery carriers of chemotherapeutic drugs and PTAs include carbon nanotubes (CNTs).” Ideally, when combining two PTAs (ICG and CNTs), the photothermal conversion ability should synergistically improve upon NIR 808 nm light exposure. This point is not discussed, which could make the synthesized final product more potent.

Reviewer 3 Report

Comments and Suggestions for Authors

Very relevant and excellent piece of work with some minor details to add to make it fully clear

1) The authors should indicate and briefly explain how they obtained the melting temperatures of DTDP and DTDPA  they report

2) In figure 2 the authors claim that all nanosystems exhibit a more rapid drug release at pH 5 but provide no clear proof. They should either elaborate more or add quantitative values of kinetic constants

3) The authors should clarify how photothermal conversion efficiencies are calculated

Comments on the Quality of English Language

minor changes

Round 2

Reviewer 1 Report

Comments and Suggestions for Authors

The authors have addressed all my queries.